# Structural Bioinformatics Applied to Acetylcholinesterase Enzyme Inhibition

**DOI:** 10.3390/ijms26083781

**Published:** 2025-04-17

**Authors:** María Fernanda Reynoso-García, Dulce E. Nicolás-Álvarez, A. Yair Tenorio-Barajas, Andrés Reyes-Chaparro

**Affiliations:** 1Departamento de Morfología, Escuela Nacional de Ciencias Biológicas, Instituto Politécnico Nacional, Unidad Profesional Lázaro Cárdenas, Prolongación de Carpio y Plan de Ayala s/n, Col. Santo Tomás, Alcaldía Miguel Hidalgo, Mexico City 11340, Mexico; ferreynoso4@gmail.com; 2Departamento de Fisiología, Escuela Nacional de Ciencias Biológicas, Instituto Politécnico Nacional, Av. Wilfrido Massieu S/N, Unidad Profesional Adolfo López Mateos, Mexico City 07738, Mexico; 3Laboratorio de Nanobiotecnologia, Facultad de Ciencias Físico Matemáticas, Benemerita Universidad de Puebla, Av. San Cladio y 18 Sur, Col. San Manuel, Edif. FM6-108, Ciudad Universitaria, Puebla 72570, Mexico; aldo.tenorio@fcfm.buap.mx

**Keywords:** acetylcholinesterase, bioinformatics, docking, virtual screening, drug design

## Abstract

Acetylcholinesterase (AChE) is a critical enzyme involved in neurotransmission by hydrolyzing acetylcholine at the synaptic cleft, making it a key target for drug discovery, particularly in the treatment of neurodegenerative disorders such as Alzheimer’s disease. Computational approaches, particularly molecular docking and molecular dynamics (MD) simulations, have become indispensable tools for identifying and optimizing AChE inhibitors by predicting ligand-binding affinities, interaction mechanisms, and conformational dynamics. This review serves as a comprehensive guide for future research on AChE using molecular docking and MD simulations. It compiles and analyzes studies conducted over the past five years, providing a critical evaluation of the most widely used computational tools, including AutoDock, AutoDock Vina, and GROMACS, which have significantly contributed to the advancement of AChE inhibitor screening. Furthermore, we identify PDB ID: 4EY7, the most frequently used AChE crystal structure in docking studies, and highlight Donepezil, a well-established reference molecule widely employed as a control in computational screening for novel inhibitors. By examining these key aspects, this review aims to enhance the accuracy and reliability of virtual screening approaches and guide researchers in selecting the most appropriate computational methodologies. The integration of docking and MD simulations not only improves hit identification and lead optimization but also provides deeper mechanistic insights into AChE–ligand interactions, contributing to the rational design of more effective AChE inhibitors.

## 1. Introduction

Acetylcholinesterase (AChE) is a key enzyme in the regulation of cholinergic neurotransmission, catalyzing the hydrolysis of acetylcholine at the synapse. Its biomedical significance lies in its role in various neurological diseases, particularly Alzheimer’s disease, where its inhibition has been the main therapeutic strategy to improve cognitive function in patients with neurodegenerative decline [1]. It is known that AChE activity can accelerate the aggregation of the β-amyloid (Aβ) peptide, increasing its neurotoxicity [2]. AChE inhibitors such as Donepezil, rivastigmine, and galantamine have shown efficacy in symptom management, although they present limitations such as side effects and a progressive loss of efficacy over time [3].

In pharmacology, AChE is also a therapeutic target for other neurological and muscular disorders, including myasthenia gravis, organophosphate poisoning, and Parkinson’s disease Recent studies have identified antiviral drugs, such as tilorone, that exhibit inhibitory activity against AChE, opening new possibilities for drug repurposing [4].

Research on AChE modulation has also explored multitarget drug design. These compounds not only inhibit AChE but also interact with other molecular targets involved in neurodegeneration, such as β-amyloid aggregation and oxidative stress [5,6,7]. This strategy aims to simultaneously address multiple pathological mechanisms, offering a potentially more effective therapeutic approach for neurodegenerative diseases. Additionally, advances in computational chemistry and molecular screening have led to the identification of new inhibitor classes with greater selectivity and lower toxicity [8,9,10]. These findings establish AChE as a target of great interest in pharmacology and biomedicine, with applications extending beyond the neurological field to new therapeutic opportunities in various diseases.

In recent years, structural bioinformatics has revolutionized the study of acetylcholinesterase (AChE) as a therapeutic target, enabling the design and optimization of novel inhibitors with greater efficacy and specificity. The availability of three-dimensional models of AChE and its interactions with different ligands has provided crucial insights into the dynamics of its active site and molecular inhibition mechanisms [11]. Using techniques such as molecular dynamics and docking studies, researchers have identified new classes of compounds capable of selectively binding to the enzyme, minimizing adverse effects, and optimizing bioavailability [12,13]. These advances have facilitated the exploration of multitarget hybrid structures, which not only inhibit AChE but also possess antioxidant or anti-inflammatory properties, opening new therapeutic avenues for neurodegenerative diseases.

One of the most significant achievements in this field has been the application of artificial intelligence and machine learning to the virtual screening of potential AChE inhibitors. Recent studies have shown that by training deep learning models with large databases of bioactive compounds, it is possible to predict with high accuracy which molecules exhibit strong affinity for the enzyme [14,15]. These techniques have significantly reduced the time and cost required for the identification of new drugs, allowing for the exploration of thousands of compounds within hours instead of years of laboratory experimentation [16,17]. Additionally, the use of computational pharmacophores and quantum simulations has enabled a more detailed characterization of key interactions between AChE and its inhibitors, enhancing the rational design of new therapies [18].

The impact of structural bioinformatics on AChE pharmacology is not limited to Alzheimer’s disease but has also extended to other biomedical applications, such as the development of antidotes against neurotoxins and organophosphate pesticides. Computational simulations have enabled the design of reversible inhibitors that can be used as rescue agents in cases of acute poisoning [19,20]. Similarly, strategies based on the chemical modification of classical inhibitors have been developed to improve their selectivity and reduce their side effects on other enzymatic systems [21,22]. Thanks to these advances, the combination of bioinformatics, computational chemistry, and structural pharmacology continues to open new possibilities for the development of safer and more effective drugs targeting AChE, reinforcing its status as a key target in precision medicine.

This article presents an analysis of recent advancements in the use of structural bioinformatics tools and computational chemistry for the design and optimization of acetylcholinesterase (AChE) inhibitors. Different approaches were reviewed, starting with the most used tools, such as molecular docking and molecular dynamics. Additionally, new approaches utilizing machine learning were considered, highlighting their impact on the identification of new compounds with greater specificity and lower toxicity. Furthermore, the role of these techniques in the development of multitarget drugs and the search for antidotes against neurotoxins is explored, consolidating the relevance of AChE as a therapeutic target in various neurodegenerative and toxicological diseases.

## 2. Acetylcholinesterase: Structure and Function

Acetylcholinesterase (AChE) is a hydrolase belonging to the cholinesterase family, whose primary function is the hydrolysis of the neurotransmitter acetylcholine (ACh) into choline and acetic acid, for allowing the termination of cholinergic signaling at the synapse [23]. It is classified within the serine esterase family due to the presence of a catalytic site characterized by the Ser–His–Glu catalytic triad, which is essential for its mechanism of action [24]. There are two main cholinesterases in higher organisms: acetylcholinesterase (AChE), which is highly specific for acetylcholine, and butyrylcholinesterase (BChE), which exhibits a more nonspecific activity and can hydrolyze other choline esters [25]. AChE plays a crucial role in both the central and peripheral nervous systems, where it regulates cholinergic transmission at neuromuscular junctions and in various brain regions, being essential for cognitive functions such as learning and memory [23].

From a structural perspective, AChE has been extensively characterized using X-ray crystallography, allowing for a high-resolution description of its three-dimensional structure [24,25]. The enzyme features a deep and narrow active site, known as the catalytic gorge, which consists of a catalytic site where substrate hydrolysis occurs, as well as a peripheral site that regulates acetylcholine entry (Figure 1) [26]. The active site comprises the Ser203–His447–Glu334 catalytic triad, where serine acts as a nucleophile in the hydrolysis reaction [27]. Additionally, the peripheral site contains aromatic residues that interact with the substrate and allosteric modulators, playing a key role in enzyme inhibition [27]. These structural features make AChE an attractive target for pharmacological drug design, particularly in the development of inhibitors for the treatment of neurodegenerative diseases [28].

The tissue and intracellular localization of AChE varies depending on its function and the cell type in which it is expressed. It is widely distributed in the central nervous system (CNS), specifically in the cerebral cortex, hippocampus, and brainstem, where it plays a role in regulating cholinergic neurotransmission [29]. In the peripheral nervous system, it is present at neuromuscular junctions, facilitating skeletal muscle contraction and relaxation. It is also expressed in erythrocytes, where it contributes to the regulation of responses to neurotoxic agents.

At the cellular level, AChE can be anchored to the plasma membrane, associated with the outer synaptic surface, or exist in soluble forms within intracellular compartments [30]. Its regulation occurs through gene expression mechanisms, post-translational modifications, and allosteric modulation, as well as interactions with proteins such as cholinergic components, neuropilins, and extracellular matrix proteins, which influence its activity and stability [31].

## 3. Structural Bioinformatics Methods in the Study of AChE

Molecular docking is a computational technique used in drug design to predict the interaction between a small molecule (ligand) and a biological macromolecule (protein or DNA) [32]. This approach is based on exploring the conformational space of the ligand and its accommodation within the active site of the target protein, evaluating binding affinity through optimization algorithms and energy functions [33]. This method allows for the identification of compounds with a high probability of specific binding, facilitating the prioritization of candidates for experimental assays. There are two main strategies in molecular docking: rigid docking, in which both the ligand and the protein maintain fixed structures, and flexible docking, which allows for the exploration of conformational changes in the ligand or protein [34]. This technique is widely used in the identification of enzyme inhibitors, the development of multitarget drugs, and the optimization of compounds in computational pharmacology.

Among the most commonly used molecular docking software, AutoDock and AutoDock Vina stand out as open-source tools widely applied in ligand–protein interaction predictions using genetic algorithms and stochastic searches [32]. Other popular programs include Molecular Operating Environment (MOE), which employs empirical energy functions to evaluate complex stability, and SwissDock, which is based on the EADock DSS engine and optimized for high-efficiency conformational exploration [35]. Additionally, Glide (Schrödinger Inc., New York, NY, USA), which incorporates quantum force models, and Gold, recognized for its accuracy in binding mode prediction, are also widely used [34]. Thanks to these tools, molecular docking has revolutionized structural pharmacology by enabling the virtual screening of thousands of compounds in significantly less time than traditional experimental methods, facilitating drug discovery and optimization.

Acetylcholinesterase (AChE) has been extensively studied using bioinformatics approaches to understand its structure, function, and interactions with potential inhibitors. These computational studies have provided valuable insights into the enzyme’s active site dynamics, facilitating the design of more effective and selective AChE inhibitors. For instance, molecular docking and molecular dynamics simulations have been employed to predict how various compounds interact with AChE, aiding in the identification of promising therapeutic candidates for neurodegenerative diseases such as Alzheimer’s disease. The continuous advancements in bioinformatics tools and techniques have significantly enhanced our ability to model AChE’s behavior and its interactions at a molecular level [36].

The importance of AChE as a pharmacological target ensures that bioinformatics studies will continue to progress in the future. As new computational methods emerge, they provide more precise modeling of AChE’s structure and its binding affinities with various ligands. These advancements not only expedite the drug discovery process but also reduce reliance on extensive laboratory experiments in the early stages of research [37]. Moreover, bioinformatics approaches enable the screening of vast chemical libraries to identify potential AChE inhibitors, thereby broadening the scope of therapeutic exploration. The integration of bioinformatics with experimental studies remains a powerful strategy for developing effective treatments targeting AChE [38].

## 4. Applications of Molecular Docking in AChE

Molecular docking has been extensively used to identify acetylcholinesterase (AChE) inhibitors and analyze the interaction of previously known compounds, aiming to develop more effective drugs for treating Alzheimer’s disease. Additionally, the detailed study of these interactions provides valuable insights into the activation and inhibition mechanisms of AChE. Understanding these processes at the molecular level not only allows for the optimization of existing inhibitors but also facilitates the design of new molecules with greater selectivity and efficacy in modulating enzymatic activity [39]. A quantitative analysis of research output reveals an increase in molecular docking studies focused on acetylcholinesterase over the past five years (Figure 2). This trend correlates with the mounting interest in multi-target ligands for neurodegenerative diseases.

A compilation of molecular docking studies was conducted in which acetylcholinesterase (AChE) was used as a pharmacological target. These studies aim to design and identify the reversible and partial-action inhibitors of AChE, which is crucial for developing new treatments, especially for neurodegenerative diseases such as Alzheimer’s. To achieve this, various strategies have been explored, including the design of new molecules through computational approaches and the evaluation of existing compounds with potential inhibitory activity. This comprehensive analysis evaluates 121 published studies investigating AChE inhibitors through molecular docking approaches, with a particular focus on three critical pharmacological parameters: selectivity, affinity, and reversibility (Appendix A Appendix A).

Among the most frequently used resources in these studies, the crystal structure 4EY7 stands out, appearing in a total of 31 records. It is followed by 4M0E with 14 records and 4EY6 with 13. The prominence of 4EY7 is notable, as this structure is co-crystallized with Donepezil, the most commonly used positive control in the assays. In contrast, 4EY6 is co-crystallized with galantamine, another widely used reference inhibitor. Most studies employ one of these inhibitors as a positive control, and in some cases, both are used. Regarding computational tools, AutoDock Vina is the most widely used software for molecular docking due to its efficiency in calculating binding energies and its ability to explore different conformations of the evaluated compounds. Moreover, it is the most widely cited open-source molecular docking software in the literature. These approaches have led to significant advancements in identifying new AChE inhibitors and optimizing compounds with therapeutic potential.

The prevalence of these studies is enabled by well-established AChE structural data, with the most frequently employed crystal structures being PDB 4EY7, 4M0E, 4EY6, 4EY5, and 1EVE. AutoDock Vina emerged as the most used tool. Docking studies incorporated subsequent molecular dynamics simulations, with simulation timescales typically ranging from 50 to 200 ns for system equilibration (Figure 3).

## 5. Molecular Dynamics Simulations in AChE

Molecular dynamics is a computational tool that allows for the study of the flexibility and temporal behavior of acetylcholinesterase (AChE) and its interactions with various inhibitors. The essence of this approach is to verify the stability of the predicted interactions obtained from molecular docking assays. However, it is a computationally demanding process that requires significant resources and simulation time. For this reason, molecular docking is often used to screen large databases and select the best candidates for subsequent testing with molecular dynamics. For example, Ref. [40] evaluated a library of 2270 phytochemicals, identifying three promising compounds that demonstrated stability in protein–ligand complexes during 100-nanosecond simulations, suggesting their potential as AChE inhibitors. Similarly, Ref. [41] designed and synthesized 18 new pyrrolidin-2-one derivatives; however, they only demonstrated the efficacy of two compounds through molecular dynamics simulations that formed stable complexes with AChE, indicating their potential effectiveness as inhibitors. A compilation of studies utilizing molecular dynamics with acetylcholinesterase and a test molecule is presented in Appendix A Appendix A. 

## 6. Discussion

Molecular docking assays have emerged as a fundamental tool in drug development, particularly in the identification of acetylcholinesterase (AChE) inhibitors. Their application has optimized the search for compounds with high specificity and affinity towards the enzyme, which is crucial in designing therapies for neurodegenerative diseases such as Alzheimer’s. AChE inhibition is a validated therapeutic strategy, as reducing the degradation of acetylcholine enhances cholinergic neurotransmission, a key aspect in mitigating the symptoms of Alzheimer’s disease [42]. However, developing effective inhibitors faces multiple challenges, including selectivity, toxicity, and the ability to cross the blood–brain barrier. In this context, molecular docking has not only facilitated the identification of new compounds with therapeutic potential but also allowed for the structural optimization of existing drugs, improving their pharmacokinetic profile and reducing adverse effects [43]. Moreover, combining docking with advanced techniques such as molecular dynamics and machine learning has enhanced its accuracy and efficiency, enabling better predictions of drug–receptor interactions [16,44]. Given the impact of these methodologies on streamlining drug discovery, it is evident that their use will continue to evolve and solidify as key tools.

Acetylcholinesterase (AChE) is characterized by an active site located at the bottom of a deep and narrow cleft approximately 20 Å deep and 5 Å wide. This cleft is lined with 14 highly conserved aromatic residues, among which tryptophan 84 (Trp84) plays a crucial role in the binding of the acetylthiocholine [45]. Inside its active site, it presents a catalytic triad formed by serine, histidine, and glutamate, located at the bottom of the cleft. This structural arrangement poses significant challenges for molecular docking assays, where the goal is to obtain ligands that interact with the catalytic triad. Nonetheless, the primary aim is to achieve reversible inhibition, so interaction with the enzyme’s peripheral site is sufficient to cause inhibition and will favor the reversibility of the inhibition [19,23]. Additionally, AChE inhibitors that act at both the active site and the peripheral anionic site (PAS) have been developed to prevent Aβ fibril aggregation [2]. Advances are aimed at finding new, highly selective, reversible, and effective inhibitors of AChE to improve treatments for neurodegenerative diseases.

AutoDock Vina is the most widely used software for molecular docking studies with the acetylcholinesterase (AChE) enzyme. Its open-source nature, efficiency, and ease of use make it ideal for screening large databases of potential compounds. Furthermore, AutoDock Vina requires minimal computational resources and offers fast processing times, facilitating its integration into drug discovery workflows. For example, screening work can be conducted using techniques that require few computational resources, and subsequently, more expensive techniques such as molecular dynamics can be employed [46].

A direct comparison of the results obtained from different docking programs is not recommended due to variations in their algorithms and scoring functions [47]. Each software may generate different binding affinity values even when using the same PDB files and ligands [48]. For instance, it has been observed that programs like CDOCKER and ClusPro yielded higher binding affinity values compared to others (Table 1). Additionally, tools like Schrödinger have shown a greater range of variability in the results, with affinity energy results ranging from −76.3 to 0.8 Kcal/mol observed in this work. To make comparisons and ana lyze our results, it is advisable to use known molecules for which experimental data or proven tests are available.

Molecular dynamics studies are a way to verify that the complexes obtained as a result of molecular docking represent stable interactions. The most widely used software for performing molecular dynamics simulations of ligand–receptor complexes, where the receptor is typically a protein, is GROMACS. GROMACS is open-source software that enables highly efficient molecular dynamics simulations, maximizing the use of available computational resources and being compatible with several affordable GPU cards on the market [49,50].

The simulation time varies depending on the phenomenon being studied in the system. Generally, for evaluating the stability of a ligand–receptor complex, 100 ns is sufficient to confirm its stability. In some cases, if the ligand moves or adjusts within the active site during the simulation, the RMSD tends to increase (>2 nm). If the ligand readjusts, the RMSD usually decreases again; however, if the RMSD remains elevated, it provides information about the ligand’s instability in the binding site where it was initially docked through molecular docking studies.

Molecular dynamics is a method to validate docking results; however, verification can also be performed through experimental studies or complemented with other computational analyses.

The integration of experimental data with molecular docking studies is essential to validate and improve the accuracy of computational predictions in identifying acetylcholinesterase (AChE) inhibitors. For example, in a recent study, benzofuran derivatives were designed and synthesized as potential AChE inhibitors. In vitro biological assays revealed that compound **7c** exhibited promising inhibitory activity, with an IC_50_ of 0.058 μM. These experimental findings were consistent with molecular docking results, which showed a favorable interaction of compound **7c** in the active site of AChE [51]. For acetylcholinesterase, the most used molecule for comparison is Donepezil, and there is variation in the resulting binding energy (Table 2); therefore, it should be considered for inclusion as a control in each study so that docking assays are conducted under the same conditions as the molecules of interest.

The appropriate selection of crystal structures from the Protein Data Bank (PDB) is essential for obtaining accurate results in molecular docking assays. For acetylcholinesterase, it was observed that the most used crystal is 4EY7, with many more citations than the other crystals (Table 3). A key consideration for selecting a PDB crystal is the resolution of the structure; higher resolutions provide more precise details about the conformation of the active site, which improves the accuracy of docking. Additionally, it is crucial to assess the presence of co-crystallized ligands or inhibitors, as these can induce conformational changes in the protein that affect its interaction with new compounds. Crystal-structure 4EY7 stems from human acetylcholinesterase, crystallized with Donepezil, and has a resolution of 2.35 Å, an acceptable value. Being crystallized with Donepezil allows for comparing inhibition by this drug and searching for similar interactions for the new molecules to be tested [95].

Across the reviewed studies, molecular docking analyses revealed consistent and critical interactions between the tested ligands and key amino acid residues within the active or peripheral binding sites of acetylcholinesterase (AChE). Notably, π–π stacking and cation–π interactions with Trp86, Trp286, and Tyr337 emerged as highly recurrent and relevant across multiple compounds, supporting their crucial role in ligand anchoring and enzymatic inhibition (e.g., Refs. [68,96,97]). Hydrogen bonding with residues such as Ser203, Glu202, Tyr124, and His447 was also frequently observed, enhancing ligand stability within the catalytic site, as reported in Refs. [77,98,99]. Compounds targeting the peripheral anionic site (PAS), like those in Refs. [93,100], often interacted with Asp74 and Tyr341, which are known to influence β-amyloid aggregation. These overlapping interactions suggest that ligands forming multiple contacts with both CAS and PAS residues—especially those involving Trp86, Tyr337, and Phe295—may exhibit superior dual-site binding and therapeutic potential for Alzheimer’s disease treatment.

Some additional considerations to keep in mind include the presence of water molecules in the active site, which are necessary if the goal is to emulate substrates, as a water molecule is required for the chemical reaction [101]. However, most studies aim to develop a molecule that inhibits reversibly, meaning that interaction with the peripheral site is sufficient to generate enzyme inhibition. Additionally, some inhibitors bind to sites other than the active site, and understanding these mechanisms will allow for the development of molecules with greater versatility to inhibit AChE to different extents and at varying concentrations [102,103].

Molecular docking will always produce a result, as the software is designed to provide the best possible ligand conformation within the protein. However, interpretation and comparison depend on a well-designed experiment, having a reference point such as Donepezil, and understanding how the protein functions.

## 7. Limitations of Computational Tools

Computational tools play a crucial role in drug design and molecular docking, offering rapid analysis and reducing the costs associated with experimental methods. However, their usage comes with notable limitations.

Energy affinity depends on the algorithms employed to calculate, and this is specified according to the software used. Another limitation is the availability of protein data; in some cases, the protein of interest is partially described, and the crystal structure is incomplete. Additionally, the metabolism does not concern, so docking only considers the energy affinity at a specific point in time [104], this suggest that we need the molecular dynamic to examine the interaction in a fine way On the other hand, molecular dynamics involves ligand–protein interaction over long periods of time, which requires deep learning and machine learning methods but also high computational power due to using quantum mechanical methods to calculate the stability and fluctuation of ligand–protein binding [105,106,107].

Computational studies or in silico studies are faster than in vivo or in vitro experiments; however, they are not substitutes. In silico studies are used to predict and estimate the interaction sites during protein–ligand binding.

## 8. Conclusions

Molecular docking has established itself as a fundamental tool in the rational design of acetylcholinesterase (AChE) inhibitors, providing valuable insights into ligand binding interactions and selectivity. However, the complexity of AChE’s active site, its deep and narrow gorge, and the involvement of key residues in ligand recognition present challenges that require a careful and systematic approach. The selection of appropriate crystal structures, such as PDB ID: 4EY7, and the inclusion of well-characterized control molecules like Donepezil, are crucial for ensuring reliable docking studies. Furthermore, docking alone may not fully capture the dynamic nature of ligand binding; thus, molecular dynamics (MD) simulations, particularly with GROMACS, have become indispensable for validating the docking results by assessing the stability and conformational flexibility of ligand–protein complexes over time. Studies indicate that a 100 ns MD simulation is often sufficient to confirm the stability of docked ligands but longer simulations may be necessary for more complex interactions. Additionally, integrating experimental validation with computational predictions enhances the accuracy and applicability of in silico methods in drug discovery. As molecular docking and MD simulations continue to evolve, their combined application will remain a cornerstone in the development of highly selective, reversible, and effective AChE inhibitors, paving the way for improved therapeutic strategies against neurodegenerative diseases.

## Figures and Tables

**Figure 1 ijms-26-03781-f001:**
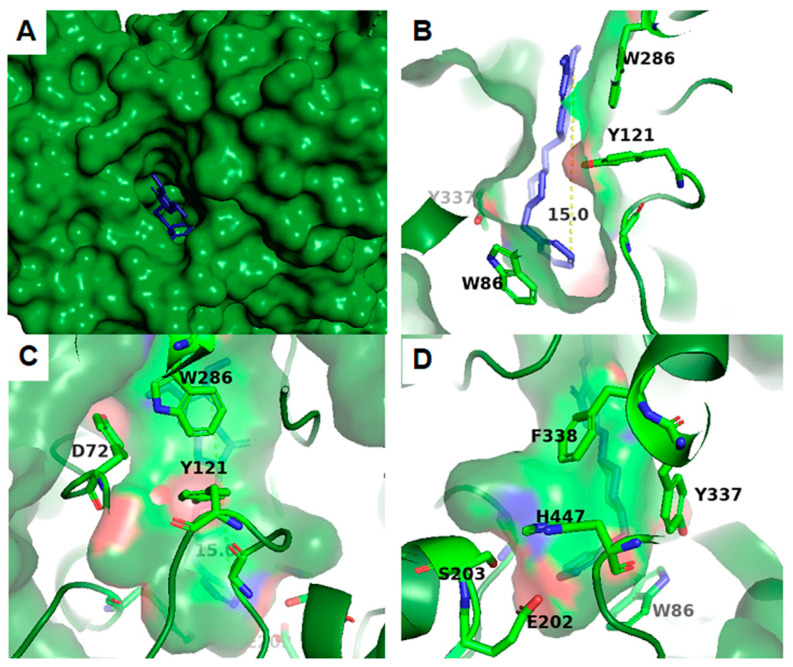
Representation of acetylcholinesterase and its active site in different visualizations. (**A**) Upper view of the pocket in AChE PDB ID: 4EY7 monomer A (in green surface) and Donepezil (in blue sticks). (**B**) Side view slice of the pocket 15 Å deep in AChE PDB ID: 4EY7 monomer A (in green surface) and Donepezil (in blue sticks). (**C**) Side view rotated 90 degrees. Three main amino acids in contact with Donepezil are shown: D72, Y121, and W286; Donepezil is represented by blue sticks and the AChE protein by green transparent surface to better observe the pocket. (**D**) Catalytic triad at the bottom of the gorge composed of H447, E202, and S203; Y337 and F338 are observed in the middle of the gorge.

**Figure 2 ijms-26-03781-f002:**
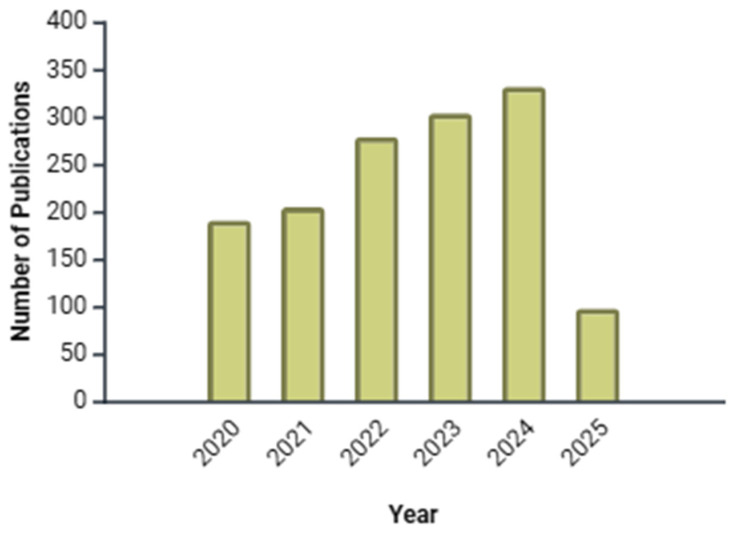
Annual publication trends in molecular docking studies related to AChE (2020–2025). The bar graph illustrates a steady increase in research output, reflecting growing interest in computational approaches for AChE inhibition and drug discovery. Data source from PubMed consulted up to March 2025.

**Figure 3 ijms-26-03781-f003:**
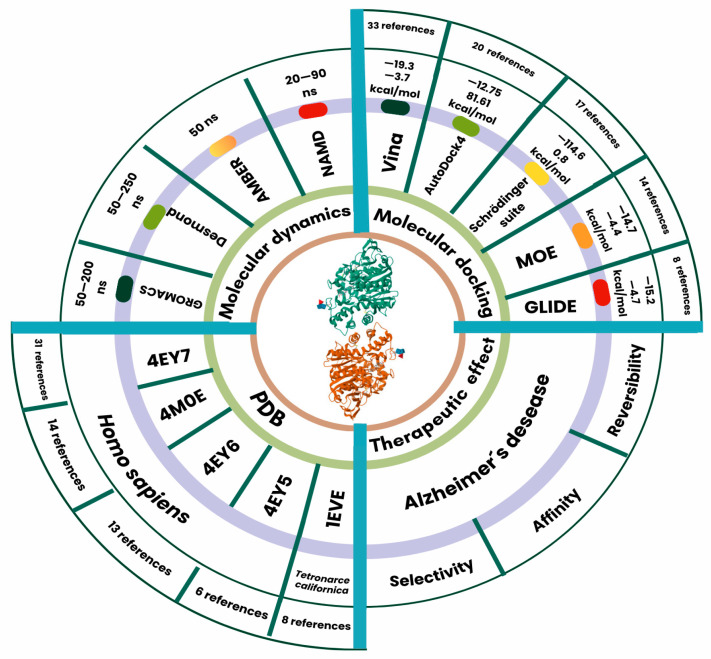
Acetylcholinesterase (AchE) as a target in docking studies. Systematic analysis of 121 Molecular Docking Studies Targeting Acetylcholinesterase (AChE) in Alzheimer’s disease therapeutics. The color-coded bar chart (green to red gradient) illustrates software usage frequency, revealing that AutoDock Vina and GROMACS emerged as the most used tools.

**Table 1 ijms-26-03781-t001:** Compilation of results from molecular docking assays, grouped by software used.

Software	No. Refs.	Binding Affinity [Kcal/mol]			
		Mean	Min	Max	STDV
Vina	33	−9.76	−19.3	−3.7	−2.45
AutoDock4	20	−7.68	−12.75	81.61	11.47
Schrödinger Suite	17	−13.40	−114.6	0.8	18.10
MOE	14	−8.25	−14.7	−4.4	2.36
Glide	8	−9.92	−15.2	−4.7	2.44
Biovia Discovery Studio	6	−13.04	−32.2	−4.6	8.40
CDOCKER	4	−3.51	−46.3	84.5	45.53
GOLD	3	14.27	−65.8	60.1	54.95
DSHC	2	−28.08	−54.5	−16.6	10.90
FlexX	3	−24.63	−36.5	−11.8	4.52
Smina	2	−10.92	−15.0	−8.6	1.70
Achilles Docking Server	1	−7.79	−8.5	−6.9	0.68
ArgusLab 4.0	1	−9.72	−12.7	−8.9	0.98
ClusPro y Pymol	1	−883.10	−974.0	−792.2	128.55
DOCK	1	−30.71	−52.4	−14.0	12.42
ICM Pro Molsoft	1	−15.79	−18.4	−12.7	1.68
iGEMDOCK	1	−87.49	−92.4	−81.1	3.73
PyVSvina	1	−10.60	−10.60	−10.60	0.0
Surflex-Dock	1	−8.96	−114.6	18.1	12.847
Vina + Umbrella Sampling simulation	1	−37.14	−53.3	−18.5	14.53

**Table 2 ijms-26-03781-t002:** Docking studies conducted with the ligand Donepezil in different research works.

Donepezil				Galantamine			
PDB	Affinity Energy (kcal/mol)	Software	Cite	PDB	Affinity Energy (kcal/mol)	Software	Cite
1ACL	−6.3226	MOE	[52]	1W6R	−9.63	AutoDock4	[53]
1C2B	−11.7	Vina	[54]		−8.68		[55]
1EVE	−12.74	ArgusLab 4.0	[56]	1DX6	−28.53	Vina	[57]
	−9.81	Glide	[58]	4EY5	−7.7	Vina	[59]
	−9.81			1C2B	−7.9	Blind docking	[60]
	−6.49	Schrödinger suite	[61]	4EY6	−7.91	AutoDock4	[62]
1OCE	−8.04	MOE	[63]		−9.9		[64]
4BDT	−8.5		[65]		−11.54	Glide	[66]
4EY5	−10.5	Vina	[59]		59.74	GOLD	[67]
	−12.42	AutoDock4	[68]		−9.28	MOE	[69]
4EY7	−10.8	Biovia Discovery studio	[70]		−7.07		[71]
4EY7	−31.26	CDOCKER	[72]		−14.2	Smina	[73]
	−5.552	Glide	[74]		−9.61	Vina	[15]
	−17.7	ICM Pro Molsoft	[75]		−9.1		[76]
	−15.5	MOE	[77]	4EY7	−9.268	Glide	[74]
	−10.171	Schrodinger suite	[78]		−9	Vina	[79]
	−18.909		[80]		−8.9		[81]
	−8.7	Vina	[79]		−10.5		[82]
	−11.94		[15]	4M0E	−21.2	FlexX	[83]
	−11.8		[84]		−7	Glide	[84]
	−10.5		[85]	5HFA	−8.2	Biovia Discovery Studio	[86]
	−11.7		[87]	6O4W	−10.4	Vina	[88]
	−18.1		[73]	6O4X	−8.02	Glide	[89]
4M0E	−45.18	CDOCKER	[90]	NA			
	−8.271	Schrödinger suite	[81]				
4PQE	−8.6	Biovia Discovery Studio	[91]				
6O4W	−14.817	Glide	[92]				
6O4X	−11.1	Vina	[93]				
73EH	−11.6		[94]				

**Table 3 ijms-26-03781-t003:** Compilation of acetylcholinesterase crystals used in molecular docking studies.

No. References	PDB ID
31	4EY7
14	4M0E
13	4EY6
8	1EVE
6	4EY5
4	4PQE, 6O4W
3	1C2B, 4BDT, 6O4X, 7D9P
2	1ACJ, 1DX6, 1OCE, 1W6R, 3LII, 4EY4, 6H12, 7D90, 7D9Q, 7XN1
1	1C2O, 1EA5, 1EEA, 1F8U, 1GQS, 1H23, 1O86, 1QON, 2ACK, 3I6M, 3I6Z, 5FPQ, 5FUM, 5HF5, 5HFA, 6CQV, 6CQZ, 6EUC, 6EYF, 6NTL, 6NTO, 6O50, 6O69, 6U37, 6WO4, 6WUZ, 6WVO, 6WVQ, 6XYU, 73EH, 7E3H

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
