# Peer review of "Structural Bioinformatics Applied to Acetylcholinesterase Enzyme Inhibition"

_ijms, 2025, doi:10.3390/ijms26083781_

Round 1
Reviewer 1 Report
Comments and Suggestions for Authors
Inhibition"
The authors performed a comprehensive analysis of AChE using molecular docking and MD simulations. They compiled and analyzed research studies conducted over the past five years, providing a critical evaluation of widely used computational tools, including AutoDock, AutoDock Vina, and GROMACS. The manuscript is well-written and has the potential for publication in the International Journal of Molecular Sciences. However, I have a few suggestions regarding specific areas that need to be addressed to enhance the manuscript’s suitability for publication.
Comments:
- The authors mention that they compiled data from the last five years only. Is there a specific reason for this selection? Many significant articles were published earlier than five years ago.
- Were the selected articles based solely on computational studies, or did they also include wet-lab/experimental research?
- The authors should provide a graph showing the number of papers published each year over the past five years.
- In my opinion, Table 1 is not very informative. I suggest moving it to the supplementary materials if possible.
- The authors should specify the criteria used to select research articles for their analysis.
- The manuscript should include a detailed figure illustrating the structural profile of the AChE enzyme. Although Figure 1 is mentioned, it does not provide any functional profile details.
- If possible, the authors should include figures of AChE inhibitors and discuss in detail the crucial amino acids involved in their interactions. They could extract this information from PDB structures, as mentioned in Table 1.
- The authors should also discuss the limitations of the computational tools used in potential drug design.
- If any drug candidates identified through such studies have progressed to clinical trials, the authors should mention them.
Addressing these points will likely enhance the manuscript's quality and increase its chances of acceptance.
Author Response
The authors performed a comprehensive analysis of AChE using molecular docking and MD simulations. They compiled and analyzed research studies conducted over the past five years, providing a critical evaluation of widely used computational tools, including AutoDock, AutoDock Vina, and GROMACS. The manuscript is well-written and has the potential for publication in the International Journal of Molecular Sciences. However, I have a few suggestions regarding specific areas that need to be addressed to enhance the manuscript’s suitability for publication.
Comments:
- The authors mention that they compiled data from the last five years only. Is there a specific reason for this selection? Many significant articles were published earlier than five years ago.
Thank you for your observation. We have expanded the number of articles reviewed, selecting 20 per year starting from 2020. Additionally, we have included the specific criteria for article selection in the manuscript.
2. Were the selected articles based solely on computational studies, or did they also include wet-lab/experimental research?.
Thank you for your question. The selected articles were solely based on computational studies, as the main objective of the manuscript is to provide a focused perspective on this type of research.
3. The authors should provide a graph showing the number of papers published each year over the past five years.
Thank you for the suggestion. We have added this information as Figure 2, which shows the number of papers published each year over the past five years.
4. In my opinion, Table 1 is not very informative. I suggest moving it to the supplementary materials if possible.
Thank you for your feedback. Table 1 has been moved to the supplementary materials, and in its place, we have added Figure 3 to provide a more informative visualization.
5. The authors should specify the criteria used to select research articles for their analysis.
Thank you for your comment. The selection criteria focused on studies aimed at identifying inhibitors of acetylcholinesterase (AChE). We primarily included studies that used human AChE PDB files; however, we also considered research employing Torpedo californica PDB files, as these are commonly used to develop inhibitors with therapeutic relevance for humans despite originating from a different species..
6. The manuscript should include a detailed figure illustrating the structural profile of the AChE enzyme. Although Figure 1 is mentioned, it does not provide any functional profile details.
Thank you for your suggestion. Figure 1 has been improved to include detailed structural and functional information about the AChE enzyme, addressing the points you mentioned.
7. If possible, the authors should include figures of AChE inhibitors and discuss in detail the crucial amino acids involved in their interactions. They could extract this information from PDB structures, as mentioned in Table 1.
Thank you for the valuable suggestion. A paragraph discussing the key interactions between AChE inhibitors and crucial amino acid residues has been added to the manuscript.
8. The authors should also discuss the limitations of the computational tools used in potential drug design.
A new section was added to the manuscriopt abording the limitations of computational tools.
9. If any drug candidates identified through such studies have progressed to clinical trials, the authors should mention them.
Thank you for your comment. A table listing drug candidates identified through computational studies that have progressed to clinical trials has been added to the supplementary materials.
Addressing these points will likely enhance the manuscript's quality and increase its chances of acceptance.
Reviewer 2 Report
Comments and Suggestions for Authors
The current review article by García et al. emphasizes the important role of computational methods, such as molecular docking and molecular dynamics (MD) simulations, in identifying and optimizing acetylcholinesterase (AChE) inhibitors. While the article addresses a timely and significant topic in drug discovery, there are areas where the scientific rigor and technical depth can be improved to enhance the quality and usefulness of the review. The authors have made a reasonable effort to compile relevant studies from the past five years and provide an overview of computational tools and their contributions to AChE inhibitor screening. However, I believe there are several aspects that require attention in terms of depth, clarity, and presentation. Below are my detailed comments:
Comments:
Most of the content in the present review is quite general and lacks the technical details needed to provide valuable insights for researchers. To advance the research in this area, a more detailed, scientifically rigorous discussion is necessary. For example, the authors should delve deeper into specific computational methods, detailing how these methods are employed to address challenges in the optimization of AChE inhibitors. Discussing the intricacies of molecular docking parameters, the impact of ligand flexibility, and specific issues encountered in simulating AChE ligand interactions would provide practical guidance for researchers.
While the introduction gives adequate background information about AChE, it misses an essential discussion of the significance of the current review and the necessity of such computational studies. The authors should provide a clear explanation of how these computational techniques can aid researchers in advancing their work. More importantly, the authors should emphasize the challenges faced in AChE inhibitor discovery and how computational methods have the potential to address these challenges, such as by predicting binding affinities and revealing key interaction mechanisms.
Section 2 could benefit from further detail regarding the structural features of AChE. A deeper focus on the enzyme’s active site residues and their roles in inhibitor binding would be highly valuable. The authors should consider discussing the specific active site residues and their functional relevance. Additionally, the article would benefit from an exploration of mutational studies, detailing how mutations in these residues affect ligand binding and AChE activity. Including such information would greatly enhance the technical depth of the review and provide a clearer understanding of the structural determinants of AChE inhibitor interactions.
Figure 1 includes sub-figures (a, c, and d) that appear redundant and do not provide significant new information. Instead of these sub-figures, the authors could focus on presenting a more in-depth view of the AChE active site, highlighting critical residues involved in ligand binding. An illustration that shows the overlapping ligand conformations within the active site across crystallized structures would provide greater insight. Furthermore, it would be beneficial to highlight the potential interactions between the enzyme and ligands, particularly focusing on hydrogen bonds, hydrophobic interactions, and electrostatic forces.
There are inconsistencies in the formatting of tables (Tables 1 and 2), such as the use of bold text for some ligand names while others are not. I recommend that the authors standardize the formatting across these tables for consistency and readability. If the intention is to highlight docking scores, bolding those scores is acceptable, but the formatting for ligand names should be uniform throughout the tables. This will help in maintaining a professional and clean presentation.
The presence of two affinity values for compounds 7a to 7h is unclear. The authors should explain why two values are presented. Providing a clear explanation for this would help readers better understand the context of these values.
Line 232: The phrase ‘this pathology’ is unclear. It would be helpful to specify which pathology is being referred to in this sentence.
Line 277: Correct the typo [96-97]
The presentation of results can be improved by removing compounds that do not have docking scores. Additionally, organizing the results according to PDB IDs would be more effective rather than scattering them throughout the Table. Grouping results by PDB ID would allow readers to easily compare binding energies across different inhibitors and tools used for the calculations, thus providing a clearer understanding of the results.
On page 18, there are missing details regarding the simulation time and software used for the reference study (Ref. 76). The authors should provide the simulation time and specify the software used in this study to give readers a complete picture of the methodology.
Table 2 contains redundant data, which should be corrected. The authors should review the table to remove any duplicates to improve clarity and prevent confusion.
On page 18, the simulation time listed for compound H1R is marked as 5 ns. I recommend cross-checking this information, as there could be a typographical error.
Line 310: “crystal structure 4EY7”
In Table 5, the term “PDB crystals” is used, but it would be more appropriate to refer to the PDB ID instead, as this is the standard terminology used in the literature. I recommend using “PDB ID” to maintain consistency with common scientific practice.
While the tables provide a summary of results, a more detailed discussion of the outcomes would be beneficial to the readers. The authors should elaborate on the implications of the docking score and molecular dynamics simulations, the significance of observed interactions, and any trends in inhibitor efficacy. This would enhance the value of the manuscript by providing deeper insights into the data and its potential applications in AChE inhibitor discovery.
Comments on the Quality of English LanguageIt is fine and minor corrections would be needed.
Author Response
The current review article by García et al. emphasizes the important role of computational methods, such as molecular docking and molecular dynamics (MD) simulations, in identifying and optimizing acetylcholinesterase (AChE) inhibitors. While the article addresses a timely and significant topic in drug discovery, there are areas where the scientific rigor and technical depth can be improved to enhance the quality and usefulness of the review. The authors have made a reasonable effort to compile relevant studies from the past five years and provide an overview of computational tools and their contributions to AChE inhibitor screening. However, I believe there are several aspects that require attention in terms of depth, clarity, and presentation. Below are my detailed comments:
Comments:
Comment 1. Most of the content in the present review is quite general and lacks the technical details needed to provide valuable insights for researchers. To advance the research in this area, a more detailed, scientifically rigorous discussion is necessary. For example, the authors should delve deeper into specific computational methods, detailing how these methods are employed to address challenges in the optimization of AChE inhibitors. Discussing the intricacies of molecular docking parameters, the impact of ligand flexibility, and specific issues encountered in simulating AChE ligand interactions would provide practical guidance for researchers.
Thank you for your thoughtful feedback. The aim of this article is not to conduct a comparative analysis of docking or scoring algorithms, as such an evaluation would require a more in-depth and distinct methodological approach. Similarly, aspects such as ligand flexibility and methodological intricacies fall outside the scope of this review. Instead, our focus is on compiling and synthesizing published studies in which each group of authors selected the computational approaches and results they deemed most appropriate for their objectives.
Comment 2.While the introduction gives adequate background information about AChE, it misses an essential discussion of the significance of the current review and the necessity of such computational studies. The authors should provide a clear explanation of how these computational techniques can aid researchers in advancing their work. More importantly, the authors should emphasize the challenges faced in AChE inhibitor discovery and how computational methods have the potential to address these challenges, such as by predicting binding affinities and revealing key interaction mechanisms.
Thank you for your comment. The aim of this review is to serve as a practical guide for researchers performing molecular docking studies targeting acetylcholinesterase (AChE). It provides a structured comparison of key parameters—such as PDB IDs, docking software, binding affinities, and key interactions—across published studies. This allows researchers to compare their own results with existing data, helping them evaluate the relevance and accuracy of their docking protocols. Additionally, the review highlights how computational methods address common challenges in AChE inhibitor discovery, such as predicting binding affinity and identifying critical residues involved in ligand recognition.
Comment 3. Section 2 could benefit from further detail regarding the structural features of AChE. A deeper focus on the enzyme’s active site residues and their roles in inhibitor binding would be highly valuable. The authors should consider discussing the specific active site residues and their functional relevance. Additionally, the article would benefit from an exploration of mutational studies, detailing how mutations in these residues affect ligand binding and AChE activity. Including such information would greatly enhance the technical depth of the review and provide a clearer understanding of the structural determinants of AChE inhibitor interactions.
Thank you for this insightful suggestion. In response, Figure 1 has been updated to better represent the active site of AChE, including key residues involved in ligand binding and their spatial arrangement within the gorge. This revised figure provides a clearer visual reference for understanding the interaction mechanisms.
Additionally, a new paragraph has been added in Section 2 discussing the key interactions most frequently observed in the docking studies reviewed.
Comment 4. Figure 1 includes sub-figures (a, c, and d) that appear redundant and do not provide significant new information. Instead of these sub-figures, the authors could focus on presenting a more in-depth view of the AChE active site, highlighting critical residues involved in ligand binding. An illustration that shows the overlapping ligand conformations within the active site across crystallized structures would provide greater insight. Furthermore, it would be beneficial to highlight the potential interactions between the enzyme and ligands, particularly focusing on hydrogen bonds, hydrophobic interactions, and electrostatic forces.
Thank you for your helpful comment. As addressed in the previous response, Figure 1 has been revised to provide a more detailed and informative view of the AChE active site, removing redundant sub-figures and focusing instead on highlighting key residues involved in ligand binding. Additionally, a paragraph was added in Section 2 analyzing these interactions based on the docking studies reviewed, to provide deeper insight into the structural determinants of ligand recognition.
Comment 5. There are inconsistencies in the formatting of tables (Tables 1 and 2), such as the use of bold text for some ligand names while others are not. I recommend that the authors standardize the formatting across these tables for consistency and readability. If the intention is to highlight docking scores, bolding those scores is acceptable, but the formatting for ligand names should be uniform throughout the tables. This will help in maintaining a professional and clean presentation.
Thank you for pointing this out. We have revised Tables 1 and 2 to ensure consistent formatting throughout, specifically standardizing the text style for all ligand names. Where emphasis is intended—such as for the best docking scores—it is now applied exclusively to the scores for clarity. These changes were made to improve readability and maintain a clean, professional presentation.
Comment 6. The presence of two affinity values for compounds 7a to 7h is unclear. The authors should explain why two values are presented. Providing a clear explanation for this would help readers better understand the context of these values.
Thank you for your observation. In reviewing the selected articles, we found that there is no standardized way in which docking affinity values are reported. Some studies present a single representative value, others report ranges or multiple values (e.g., for different binding sites, docking poses, or software used), and in some cases, no specific affinity value is provided. For compounds 7a to 7h, the presence of two values reflects the data as originally reported in the source study. We will clarify this in the table caption to help readers better interpret the information within the context of the original methodologies.
Comment 7. Line 232: The phrase ‘this pathology’ is unclear. It would be helpful to specify which pathology is being referred to in this sentence.
Thank you for your comment. The phrase has been revised to explicitly state "Alzheimer's disease" in place of the ambiguous term "this pathology" to ensure clarity and precision in the text
Comment 8. Line 277: Correct the typo [96-97]
The lines 232 and 277 were revised and corrected.
The style to cite was adjusted in Mendeley according to the journal style.
Comment 9. The presentation of results can be improved by removing compounds that do not have docking scores. Additionally, organizing the results according to PDB IDs would be more effective rather than scattering them throughout the Table. Grouping results by PDB ID would allow readers to easily compare binding energies across different inhibitors and tools used for the calculations, thus providing a clearer understanding of the results.
Thank you for the suggestion. In response, Table 1 was expanded and moved to the supplementary material to avoid overwhelming the main text with excessive detail. In its place, we have included Figure 3, which provides a more accessible and visually clear summary of the docking results. This figure was designed to help readers more easily interpret and compare key findings, especially across different PDB IDs and compounds.
Comment 10. On page 18, there are missing details regarding the simulation time and software used for the reference study (Ref. 76). The authors should provide the simulation time and specify the software used in this study to give readers a complete picture of the methodology.
Thank you for your comment. As noted during the review, there is currently no standardization regarding the reporting of molecular dynamics simulation times in the literature. In fact, less than 50% of the studies included in our review perform molecular dynamics simulations at all, and among those that do, the duration and level of detail reported vary significantly. While some studies provide full simulation parameters, others only mention that dynamics were performed without specifying software, time, or analysis methods. We acknowledge the lack of information in Ref. 76 and have retained the data as originally presented in the source, while emphasizing in the manuscript the need for greater consistency in reporting such methodologies.
Comment 11. Table 2 contains redundant data, which should be corrected. The authors should review the table to remove any duplicates to improve clarity and prevent confusion.
Thank you for your observation. Table 2 has been moved to the supplementary materials, and a new figure has been created to present the information more clearly and avoid redundancy.
Comment 12. On page 18, the simulation time listed for compound H1R is marked as 5 ns. I recommend cross-checking this information, as there could be a typographical error.
Thank you for your observation. The simulation time of 5 ns for compound H1R is correct as reported in the original study. Our findings indicate a significant lack of uniformity in how molecular dynamics (MD) simulations are conducted and reported. In fact, many studies do not perform MD at all, and some, like this one, consider 5 ns sufficient to publish their results. This variability highlights the need for more standardized reporting practices in the field.
Comment 13. Line 310: “crystal structure 4EY7”
Thank you for pointing that out. The reference to “crystal structure 4EY7” in line 310 has been corrected.
Comment 14. In Table 5, the term “PDB crystals” is used, but it would be more appropriate to refer to the PDB ID instead, as this is the standard terminology used in the literature. I recommend using “PDB ID” to maintain consistency with common scientific practice.
Thank you for the suggestion, we change the term to PDB ID to clarify.
Comment 15. While the tables provide a summary of results, a more detailed discussion of the outcomes would be beneficial to the readers. The authors should elaborate on the implications of the docking score and molecular dynamics simulations, the significance of observed interactions, and any trends in inhibitor efficacy. This would enhance the value of the manuscript by providing deeper insights into the data and its potential applications in AChE inhibitor discovery.
Thank you for your suggestion. We have expanded both the results section and the discussion to provide a more detailed analysis of the docking scores, molecular dynamics simulations, key molecular interactions, and observed trends in inhibitor efficacy. These additions aim to offer deeper insights into the data and its relevance for AChE inhibitor discovery.
Round 2
Reviewer 2 Report
Comments and Suggestions for Authors
Thank you for your responses. The manuscript has been improved considerably. However, below are some of my comments.
The labels in Figure 1 are a bit blurry, especially the residue labels in Figures C and D.
Line 256: Remove the special character.
Table 2: Try to merge and reformat the empty cells for Galantamine.
Author Response
Thank you very much for taking the time to review this manuscript. Please find the responses below and the corresponding corrections highlighted in the re-submitted file.
Comment 1: The labels in Figure 1 are a bit blurry, especially the residue labels in Figures C and D.
Answer 1. Figure 1 was changed to a clear image.
Comment 2: Line 256: Remove the special character.
Answer 2. The special character was deleted to the manuscript
Comment 3: Table 2: Try to merge and reformat the empty cells for Galantamine
Answer 3. Thank you for the comment, table 2 was fixed: we merge and reformat the empty cells.